# Congenital Sensorineural Hearing Loss and Inborn Pigmentary Disorders: First Report of Multilocus Syndrome in Piebaldism

**DOI:** 10.3390/medicina55070345

**Published:** 2019-07-07

**Authors:** Laura Cristina Gironi, Enrico Colombo, Alfredo Brusco, Enrico Grosso, Valeria Giorgia Naretto, Andrea Guala, Eleonora Di Gregorio, Andrea Zonta, Francesca Zottarelli, Barbara Pasini, Paola Savoia

**Affiliations:** 1Department of Health Sciences, Amedeo Avogadro University of Eastern Piedmont, 28100 Novara, Italy; 2Department of Translational Medicine, Amedeo Avogadro University of Eastern Piedmont, 28100 Novara, Italy; 3Department of Medical Sciences, University of Turin, 10124 Torino, Italy; 4Medical Genetics Unit, Città della Salute e della Scienza University Hospital, 10124 Torino, Italy; 5Maternal Infant Department, Castelli Hospital, 28922 Verbania, Italy

**Keywords:** genodermatoses, genetic skin disorders, auditory-pigmentary disorders, neurocristopathies, piebaldism, multilocus syndrome, KIT, 15q13.3 deletion

## Abstract

Congenital sensorineural hearing loss may occur in association with inborn pigmentary defects of the iris, hair, and skin. These conditions, named auditory-pigmentary disorders (APDs), represent extremely heterogeneous hereditary diseases, including Waardenburg syndromes, oculocutaneous albinism, Tietz syndrome, and piebaldism. APDs are part of the neurocristopathies, a group of congenital multisystem disorders caused by an altered development of the neural crest cells, multipotent progenitors of a wide variety of different lineages, including those differentiating into peripheral nervous system glial cells and melanocytes. We report on clinical and genetic findings of two monozygotic twins from a large Albanian family who showed a complex phenotype featured by sensorineural congenital deafness, severe neuropsychiatric impairment, and inborn pigmentary defects of hair and skin. The genetic analyzes identified, in both probands, an unreported co-occurrence of a new heterozygous germline pathogenic variant (c.2484 + 5G > T splicing mutation) in the *KIT* gene, consistent with the diagnosis of piebaldism, and a heterozygous deletion at chromosome 15q13.3, responsible for the neuropsychiatric impairment. This case represents the first worldwide report of dual locus inherited syndrome in piebald patients affected by a complex auditory-pigmentary multisystem phenotype. Here we also synthesize the clinical and genetic findings of all known neurocristopathies characterized by a hypopigmentary congenital disorder.

## 1. Introduction

Neurocristopathies (NCP) represent a complex group of inborn conditions associated with a wide range of multiorgan congenital diseases determined by an aberration in growth, migration, and differentiation of neural crest (NC) cells [1,2]. NC is a transient, multipotent cell population that generates a broad spectrum of numerous cell types, including melanocytes of the skin and hair follicles, neurons, and glial cells. As a result, patients with NCP develop an extensive range of multisystem abnormalities such as skin and hair pigmentation defects, neurological and ocular disorders [1,2,3]. Among NCP, Hirschsprung, DiGeorge, Treacher-Collins, CHARGE, Axenfeld-Rieger, and Goldenhar syndromes are mainly featured by cardiac, auditory, eye, neurological, urinary, craniofacial defects, and/or missing enteric ganglia [1,2]. Waardenburg syndrome (WS), oculocutaneous albinism (OCA), Tietz syndrome (TS) and piebaldism (PBT) represent a subgroup of NCP, termed auditory-pigmentary disorders (APDs). All APDs are clinically characterized by inborn skin and hair pigmentary anomalies. They can also be associated with a wide and heterogeneous spectrum of auditory, neurological, and ocular disorders [4,5,6]. Precisely, congenital sensorineural hearing loss (SNHL) is a very common feature in WS and TS patients; conversely, it has been described more rarely in PBT and OCA subjects [4,5,6]. Table 1 synthesizes the clinical and genetic findings of all known neural crest–associated diseases characterized by hypopigmentary congenital disorders [3,4,5,6,7,8,9,10,11,12,13,14,15].

Here, we describe clinical features and genetic data of two Caucasian monozygotic twins affected by an unreported neuro-cutaneous phenotype characterized by inborn SNHL, a congenital disorder of pigmentation and severe neuropsychiatric impairment.

## 2. Case Presentation

### 2.1. Clinical Findings

The probands, two 21 year old male monozygotic twins (IV-3 and IV-4, Figure 1), were referred to our department for dermatologic evaluation. Clinical examination revealed numerous regularly shaped hypopigmented patches on the face, trunk, and upper and lower extremities, mainly distributed on the frontal side of the body (Figure 2). Several hyperpigmented dots and macules were present within and along the margins of the leukoderma areas. Both probands had blond hair with tufts of leukotrichia on the scalp, eyelids, and eyebrows. Both had blue eyes, but only IV-4 showed heterochromia at the right iris. They did not have *dystopia canthorum*, nor were they dysmorphic.

Patients suffered from profound bilateral congenital SNHL, neurodevelopmental delay, severe intellectual disability, and childhood onset of grafted psychosis. They displayed speech disorders and social isolation.

Probands were the first children of an unrelated couple of Albanian parents, coming from a small community of central Albania. The mother (Figure 1, III-6), a 50 year old woman, reported that her sons’ depigmented lesions were congenital and did not evolve. Her dermatological examination also showed inborn hypopigmented skin and hair lesions, similar but less extended than those of her children (Figure 2). None of her family members presented any neurological sign.

We could not perform a dermatological examination on the father, (Figure 1, III-5), a 55 year old man affected by a severe form of adult-onset schizophrenia. His wife reported he did not present any skin or hair anomalies. On the other hand, several males and female relatives of III-6 (Figure 1) presented similar skin and hair congenital lesions shown in the probands. A dermatological exam of the probands’ maternal uncle (III-9) and his 6 month old daughter (IV-6), showed diffuse areas of leukoderma and leukotrichia in the absence of neurological symptoms.

Clinical skin and hair features of the probands suggested a diagnosis of PBT, generally associated with mutations in the *KIT* gene (MIM *164920, NM_000222).

### 2.2. Materials and Methods

Genomic DNA was isolated from peripheral blood using a standard procedure (Qiagen, Hilden, Germany) and quantified by Nanodrop spectrophotometer (Thermo Scientific Waltham, Massachusetts, USA). The *KIT* gene (NM_000222.2) coding exons were amplified and sequenced using Sanger sequencing.

We performed array-CGH with a 60K whole-genome oligonucleotide microarray following the manufacturer’s protocol (Agilent Technologies, Santa Clara, California, USA). Slides were scanned using a G2565BA scanner and analyzed using Agilent CGH Analytics software ver. 4.0.81 (Agilent Technologies) with the statistical algorithm ADM-2 and a sensitivity threshold of 6.0. At least three consecutive aberrant probes identified significant copy-number changes. We compared our findings to known CNVs listed in the Database of Genomic Variants (DGV, http://projects.tcag.ca/variation) and in the DECIPHER database (https://decipher.sanger.ac.uk/).

TaqMan real-time quantitative PCR (qPCR) analysis was used to measure copy number variants at 3q23 and 15q13.2q13.3 in genomic DNA as follows: (a) 3q23 duplication, *GK5* (NM_001039547.2) exon 16, primers 5′-agactggaagctccctgaaa; 5′-tcccacatacatgaaagcaca; #38 UPL probe (Roche Diagnostics); (b) 15q13.2q13.3 deletion, *CHRNA7* (NM_000746) exon 2, primers 5′-caatgactcgcaaccactca; 5′-atccacgtccatgatctgc; #7 UPL probe (Roche Diagnostics); and (c) *RNaseP* reference gene, VIC-labeled pre-designed TaqMan gene expression assays (P/N 4316844, Applied Biosystems). We carried out the reaction with an ABI 7500 Fast real-time PCR machine using the ABI TaqMan Universal PCR master mix according to the manufacturer’s instructions (Applied Biosystems, Foster City, USA). Efficiencies of the assays were similar and in a range of 90% to 110%. Samples from affected individuals and unrelated healthy controls were run in triplicate. The mean Ct value was used for calculations using the ΔΔCt method [16].

### 2.3. Genetic Findings

The sequence of the coding exons and flanking intron sequences of the *KIT* gene allowed the identification of a heterozygous c.2484+5G>T change in intron 17. The variant was not reported in the Genome Aggregation Database (GnomAD; http://gnomad.broadinstitute.org/), and it was predicted to reduce the score of the donor splice site of exon 17 (MutationTaster, http://www.mutationtaster.org/: pathogenic; Splice Site Prediction by Neural Network, http://www.fruitfly.org/seq_tools/splice.html: from 0.99 to 0.35; transcript-inferred pathogenicity score doi:10.1038/s41467-017-00141-2: 0.93, pathogenic; CADD score = 22.4, representing <1% of the most pathogenic mutations; Human Splicing Finder 3.1 = donor splice site broken wild type vs. mutant 86.87 to 74.56, −14.17%). The Human Splicing Finder 3.1 software also predicts the complete loss of an SRp55 splicing enhancer binding site, further corroborating a deleterious effect of the c.2484 + 5G > T change on splicing. Although cDNA was not available to verify the c.2484 + 5G > T effect on splicing, this variant was classified as likely pathogenic.

Because the germline mutations in the *KIT* gene are infrequently associated with neurodevelopmental abnormalities, we performed an array-CGH analysis, which identified in IV-3 a 207-kb VUS (variant of unknown significance) duplication at the long arm of chromosome 3, and a 1.5-Mb deletion at the long arm of a chromosome 15: arr[GRCh37] 3q23(141,841,034-142,048,481) × 3, 15q13.2q13.3(31,014,508-32,510,863) × 1. The latter completely overlaps with the 15q13.3 microdeletion critical region (MIM 612001). Both rearrangements were validated in the probands by quantitative real-time PCR. The variants were excluded in both the mother and the unaffected brother.

## 3. Discussion

PBT is a rare autosomal dominant genodermatosis caused by mutations in the *c-kit* proto-oncogene, which encodes the transmembrane receptor tyrosine kinase for mast cell growth factor (MGF, also known as stem cell factor) [9,10,11,12,13,14]. The KIT receptor and its ligand (KITLG) act as crucial factors in the control of physiological and pathological skin pigmentation through the Ras ⁄ mitogen-activated protein kinase (MAPK) signaling pathway. Accordingly, loss-of-function *KIT* mutations determine defects in the survival, proliferation, differentiation, and migration of melanoblasts from the NC to the skin during early embryonic development. Consequently, patients have a significant or complete loss of melanocytes in the affected areas of the hair and skin [9,10]. PBT is clinically characterized by congenital leukoderma, leukotrichia of hair, eyebrows and eyelashes, and, rarely, heterochromia of irides, with a great variation in the degree and pattern of presentation, even within affected families. Specifically, piebald patients are featured by congenital, well-demarked, symmetrical, non-pigmented white patches involving the skin of the face, trunk, arms, and legs [6,9,10]. They also frequently show poliosis, traditionally known as “white forelock”, a localized patch of white hair in a group of hair follicle; it is often triangular in shape and may be the only manifestation of PBT in 80% to 90% of *c-Kit* carriers. In some cases, both the hair and the underlying forehead may be affected [6,9]. The skin lesions, histologically characterized by the congenital absence of melanocytes, are usually stable during life, although hyperpigmented dots or macules may appear within or at their margins. Sometimes, café-au-lait macules can be present in piebald patients who concomitantly may also be affected by neurofibromatosis type 1 (NF1, MIM *613113) [9,11]. Very few piebald subjects harbor a heterozygous change in the *SNAI2* gene. It is located on chromosome 8 at position 11.21 and encodes for SNAIL2, a protein that belongs to the Snail family of zinc finger transcription factors. SNAIL2 is involved in the regulation of differentiation and migration of NC cells during embryonic development [4,5,6]. It is interesting to note that *SNAI2* germline mutations in a homozygous state were reported in a few human cases affected by a Waardenburg syndrome type 2 [7].

We describe a large family of Albanian descent harboring a germline pathogenic change in the *KIT* gene (Figure 1). Three subjects carrying the c.2484 + 5G > T splicing mutation presented skin and hair manifestations consistent with the diagnosis of PBT and several other maternal relatives had signs consistent with PBT. Particularly, only the probands manifested SNHL and severe intellectual disability, which have been reported rarely in piebald patients. The first descriptions of PBT associated with sensorineural deafness date back to the late sixties, however, these reports lack precise genetic data. Subsequently Spritz et al. [12] described a South African female affected by severe SNHL and PBT carrying a heterozygous missense change (p.R796G) in the *KIT* gene. Human homozygosity for *KIT* germline mutations has been reported in a severe multisystem phenotype consisting of hypopigmented skin and hair, blue irides, neurodevelopmental delay, hypotonia, SNHL, anemia, brachycephaly and clinodactyly [13,14]. In view of the uniqueness of the probands’ phenotype and the rarity of neurological manifestations in PBT, we hypothesized the presence of other genetic changes.

In effect, we also found in both probands a heterozygous deletion at chromosome 15q13.3, overlapping to the 15q13.3 microdeletion syndrome minimal region (MIM #612001). This genomic disorder has been associated with variable neurological and behavioral symptoms, including cognitive impairment, epilepsy, deficits in social interaction, decreased attention spans, aggressive behaviors, autism, schizophrenia, and bipolar disorder [17,18,19]. Heterozygous deletion of chromosome 15q13.3 syndrome has incomplete penetrance and up to 75% of the described affected subjects inherited the variant from a parent with at least one of the neurodevelopmental or neuropsychiatric signs or an apparently normal phenotype [17,18,19]. The family history might be compatible with this syndrome because the deletion is likely inherited from the father, affected by severe schizophrenia. Interestingly, a paternal uncle was reported to be affected by depression.

As regards SNHL, it has never been described in association with the 15q13.3 microdeletion syndrome, despite its broad phenotypic spectrum. In conclusion, even though the presence of a third unidentified (probably autosomal recessive) genetic determinant cannot be excluded with certainty, our piebald probands could represent the second description of SNHL in carriers of a heterozygous germline mutation in *KIT* gene.

## 4. Conclusions

We describe a novel example of a complex disease in two twins, tracked back to at least two different autosomal dominant diseases which are maternally inherited PBT associated with *KIT* mutation and a 15q13.3 genomic disorder, likely inherited from the father, responsible for a neurodevelopmental disease with variable expressivity/incomplete penetrance.

To our best knowledge, this is the first reported phenotype due to the co-occurrence of germline pathogenic change in the *KIT* gene mutation and deletion at chromosome 15q13.3.

This report follows up a recent finding on large cohorts of complex patients showing 5% of cases having two or more genetic diseases. Our experience highlights the crucial role of genetic analyzes in the case of a multisystem phenotype, where the presence of multilocus syndromes should be considered [20,21]. Since two or more genetic disorders may overlap in a single individual, combining an extremely polymorphous multisystem phenotype, the diagnostic process should involve an interdisciplinary group of clinicians.

## Figures and Tables

**Figure 1 medicina-55-00345-f001:**
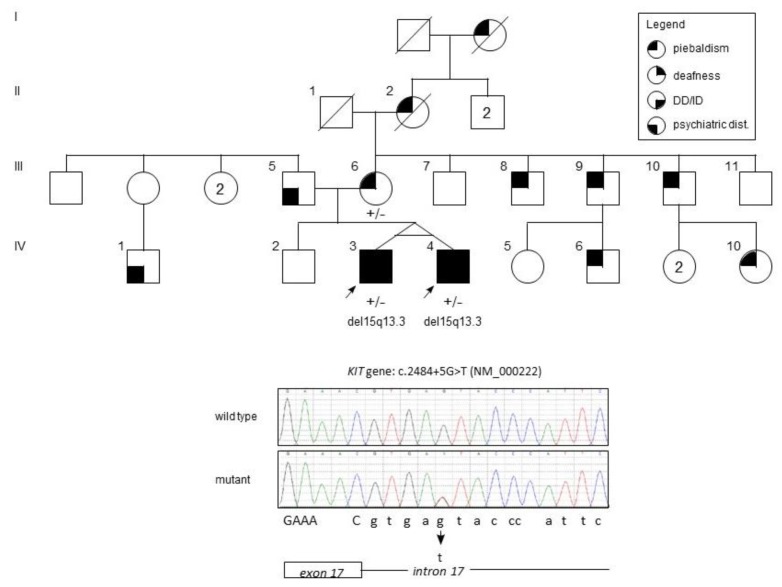
Family pedigree and sequence analysis. Top: the genealogical tree of the two probands (monozygotic twins). Squares represent male subjects, circles represent female subjects. The numbers inside symbols indicate the number of subjects with the same sex. A transversal line indicates dead subjects. The legend box indicates the features associated with each filled quadrant of the symbol. Arrows indicate the probands. DD and ID are developmental disabilities and intellectual disability, respectively. The symbol ”+/–“ indicates a heterozygous state for the c.2484+5G > T *KIT* variant. Bottom: Sanger sequences of the variant in a wild type and the IV-3 proband. Below we indicate the exon 17 last bases (capital letters) and the intron 17 region (lower case letters).

**Figure 2 medicina-55-00345-f002:**
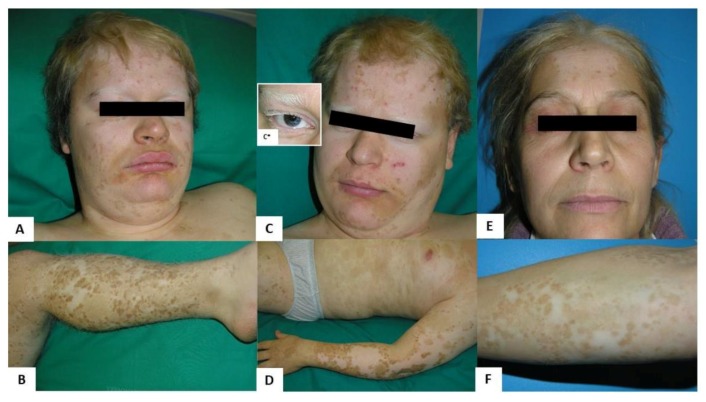
Dermatological features of probands (**A**,**B**. **IV-3**; **C**,**D**. **IV-4**) and their mother (**E**,**F**. **III-6**). All subjects presented well-demarcated, congenital, hypopigmented patches involving the face, trunk, upper and lower limbs. Hyperpigmented macules were also present mainly on the periphery and within the hypopigmented skin lesions. Poliosis, a localized patch of white hair, was present in all subjects (**A**,**C**,**E**); leukotrichia of eyebrows and eyelashes were noted only in probands (**A**, **IV-3** and **C**, IV-4). Both probands showed blue eyes, but only IV-4 had heterochromia at the right iris (**C***).

**Table 1 medicina-55-00345-t001:** Neural crest–associated diseases with hypopigmentary congenital disorders.

Name of Syndrome	Genetic Disorder ^1^	Frequency	Clinical Findings ^2^	
			Skin and Cutaneous Annexes	Non-Cutaneous
**Waardenburg Syndrome, Type 1**	**WS1 AD Gene *PAX3* (2q36.1)**	Incidence 1:20,000–40,000 births (3–5% of congenital SNHL)	**Poliosis** (45%) **Early graying of the scalp hair** (45%) **Congenital leukoderma** (30%): white skin patches on the face, trunk, or limbs; hyperpigmented borders may be present	**Facial dysmorphism:** Dystopia canthorum (100%): lateral displacement of the inner canthi, high nasal root (50–100%), medial eyebrow flare (60–70%) **Eyes:** Heterochromia iridium (15–30%), hypoplastic or brilliant blue irides (15%) **Neurological signs:** SNHL (60%) **Occasional findings:** spina bifida, cleft lip and palate
**Waardenburg Syndrome, Type 2**	**WS2** AD Genes ***MITF*** 15–20% (3p14-p13) ***SOX10*** 15% (22q13.1) **SNAI2** very rare (8q11.21)		**Poliosis** (15–30%) **Early graying of the scalp hair** (30%) **Congenital leukoderma** (5–12%)	**Facial dysmorphism:** Dystopia canthorum *not present* (0%), high nasal root (0–14%), medial eyebrow flare (10%) **Neurological signs:** SNHL (80–90%) **Eyes:** Heterochromia iridium/(50%), hypoplastic blue irides (3–23%) **Kallmann syndrome** (anosmia, hypogonadism) in carriers of mutation in *SOX10* gene **Occasional findings:** temporal bone abnormalities, ganglionic megacolon, abnormality of the kidney and/or the pulmonary artery, ptosis, telecanthus
**Waardenburg** **syndrome, Type 3** **(Klein–Waardenburg syndrome)**	**WS3** AD/AR Gene ***PAX3*** (2q36.1) The rarest form of all WS types		Hypopigmentation abnormalities of hair and skin like WS1	**Facial dysmorphism:** like WS1 **Eyes:** like WS1, blepharophimosis (80–100%) **Neurological signs:** SNHL, microcephaly (80–100%) **Limb anomalies:** hypoplasia of the musculoskeletal system, flexion contractures, carpal bone fusion, syndactyly (80–100%)
**Waardenburg syndrome, Type 4** **(Waardenburg-Shah syndrome)**	**WS4** AR Genes ***EDNRB*** (13q22.3) ***EDN3*** (20q13.32) ***SOX10*** (22q13.1) Prevalence: <1/10^6^		Hypopigmentation abnormalities of hair and skin like WS1	**Facial dysmorphism:** like WS1 **Eyes:** like WS1 **Neurological signs:** SNHL, Hirschsprung disease **Neurologic Waardenburg-Shah syndrome or PCWH** (Peripheral demyelinating neuropathy, central dysmyelinating leukodystrophy, Waardenburg syndrome and Hirschsprung disease). Association of the features of WS4 and neurological impairment (neonatal hypotonia, intellectual deficit, nystagmus, progressive spasticity, ataxia, and epilepsy)
**Tietz syndrome**	**TS** AD dominant negative effect Gene ***MITF*** (3p14-p13)	Extremely rare (only few families described)	**Generalized uniform hypopigmentation of the skin** (100%): the affected patients are born “snow white”, then they gradually gain some pigmentation (fair skin) and they may have reddish freckles **Hair, eyebrows and eyelashes** (100%): blonde to white	**Facial dysmorphism:** not present **Neurological signs:** severe congenital bilateral SNHL (100%) **Eyes:** blue eyes and hypopigmented fundi
**Piebaldism**	**PBT** AD Genes ***KIT*** (4q12) ***SNAI2*** (8q11.21)	Incidence <1:20,000	**Congenital well-demarked, symmetrical, non-pigmented patches involving the skin of the face, trunk, arms and legs.** The skin lesions are usually stable during life, although hyperpigmented dots or macules may appear within or at their margins Café-au-lait macules can be present (co-occurrence with neurofibromatosis type 1). **Poliosis** is traditionally known as “white forelock” (localized patch of white hair in a group of hair follicles). It is often triangular and may be the only manifestation of PBT in 80–90% of c-Kit carriers. **Leukotrichia of eyebrows and eyelashes**	Heterochromia irides, neurological impairment and SNHL occur rarely. Spritz et al. described a South African female affected by severe SNHL and PBT, carrying a heterozygous missense change (p.R796G) in the *KIT* gene [12] Human homozygosity for the *KIT* germline mutations has been reported in a severe multisystem phenotype consisting of hypopigmented skin and hair, blue irides, neurodevelopmental delay, hypotonia, SNHL, anemia, brachycephaly, and clinodactyly [13,14]
**Oculocutaneous** **Albinism** **OCA**	*OCA* is a heterogeneous and autosomal recessive disorder		NOTE: All types of OCA are associated with increased risk of precancerous skin lesions and skin tumors (non-melanoma and melanoma skin cancers)	NOTE: Congenital SNHL has been described in a few OCA patients as a consequence of the co-occurrence of AR deafness and OCA
	**OCA1A, OCA1B** AR Gene ***TYR*** (11q14.3)	The most common subtype in Caucasians, accounting for about 50% of cases worldwide Incidence 1:20,000/40,000 births	**Generalized congenital hypopigmentation of the skin** (white or very light) **White or nearly white or light yellow/blonde hair, eyebrows and eyelashes**	**Ocular signs:** nystagmus; reduced iris pigment with iris translucency; reduced retinal pigment with visualization of the choroidal blood vessels on ophthalmoscopic examination; foveal hypoplasia with reduction in visual acuity, strabismus, blue irides that darken to green/hazel or light brown/tan with age
	**OCA2**, AR Gene ***OCA2*** (15q12)	The most common OCA type in Africa accounting for 30% of cases worldwide Incidence 1:20,000/40.000 births	**Generalized congenital hypopigmentation of the skin** (never white but range from very fair to near normal). The skin color may darken over time and sun exposure **Lightly pigmented hair, eyebrows and eyelashes** (never white but range from light yellow to blonde to brown), the hair color may darken with age	**Ocular signs:** like OCA1 with visual acuity usually better than OCA1 **Iris color:** it ranges from blue to brown
	**OCA3**, AR Gene ***TYRP1*** (9p23)	Prevalence: 1/8.500 individuals in African population It is extremely rare in Caucasian population	**Generalized light or freckled or light brown skin** **Ginger-red or blonde hair**	**Ocular signs:** like OCA1 **Iris color:** blue or brown
	**OCA4**, AR Gene ***SLC45A2*** (5p13.3) **OCA6**, AR ***SLC24A2*** (9p22.1) **OCA5**, AR **Gene unknown** (4q24) **OCA7**, AR Gene ***C10ORF11*** (10q22.2)	Extremely rare	**Generalized congenital hypopigmentation of the skin** (never white, range from creamy white to near normal) **Lightly pigmented hair,** eyebrows and eyelashes (never white but range from silvery to golden or near normal) The hair color may darken with age.	**Ocular signs:** like OCA1 with visual acuity usually better than OCA1. OCA6 and 7 patients do not present an obvious change in the pigmentation patterns. **Iris color:** it ranges from blue to brown

Note ^1^: AD autosomal dominant, AR autosomal recessive. Note ^2^: the average penetrance of clinical signs is specified in brackets.

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
