# Peer review of "Congenital Sensorineural Hearing Loss and Inborn Pigmentary Disorders: First Report of Multilocus Syndrome in Piebaldism"

_medicina, 2019, doi:10.3390/medicina55070345_

Round 1
Reviewer 1 Report
Comments to Authors:
This manuscript is entitled “Congenital sensorineural hearing loss and inborn pigmentary disorders: First report of multilocus syndrome in piebaldism” and it is a case report of piebaldism with locus heterogeneity. While the manuscript is well written and interesting to read, I have some amendments/comments for improvement.
1- Page 5, line 120: it is a variant of uncertain (or unknown) significance (VUS); not VOUS (Variant Of Unknown Significance).
2- The authors applied Sanger sequencing of the KIT gene and array-CGH to perform the genetic testing for the patients. I wonder if they did whole exome sequencing (WES) as well and if not why they did not do it to exclude all other possible causative loci. Their reference 20th is on clinical application of the whole exome sequencing (WES).
3- According to the Human Protein Atlas database, there is a relatively high expression of KIT gene in skin. Why the authors did not confirm the effect of non-canonical splice site mutation of c.2484+5G>T in patient’s skin biopsies by RT-PCR or RNA-Seq (whole transcriptome study)?
4- In silico prediction tools are helpful to classify the variant of unknown significance (VUS) and it is better to have the results of different prediction programs. Therefore, it would be better to add the Combined Annotation-Dependent Depletion (CADD) score of the c.2484+5G>T mutation as well, that is 22.4 for this mutation. CADD is a novel and comprehensive functional annotation tool that the score of 15 can be a cutoff on deleteriousness of a variant. In addition, they can report in silico prediction metrics of the pathogenicity of c.2484+5G>T splice site mutation in Human Splicing Finder (HFS) database.

Author Response
We would be honored to resubmit to Medicina the revised manuscript entitled “Congenital sensorineural hearing loss and inborn pigmentary disorders: First report of multilocus syndrome in piebaldism”
Hereinafter, we provide a point-by-point response letter.
English language and style are fine/minor spell check required
Response: We changed the text as you suggested. English language and style of the manuscript has been completely revised. We have corrected all grammar and spelling errors.
This manuscript is entitled “Congenital sensorineural hearing loss and inborn pigmentary disorders: First report of multilocus syndrome in piebaldism” and it is a case report of piebaldism with locus heterogeneity. While the manuscript is well written and interesting to read, I have some amendments/comments for improvement.
1- Page 5, line 120: it is a variant of uncertain (or unknown) significance (VUS); not VOUS (Variant Of Unknown Significance).
Response 1: We have edited as requested.
2 - The authors applied Sanger sequencing of the KIT gene and array-CGH to perform the genetic testing for the patients. I wonder if they did whole exome sequencing (WES) as well and if not why they did not do it to exclude all other possible causative loci. Their reference 20th is on clinical application of the whole exome sequencing (WES).
Response 2: The identification of the KIT mutation and the 15q13.2q13.3 microdeletion seemed to explain well the combined phenotype identified in our patient. It must also be considered that this patient was referred to our hospital several years ago when WES was only at its beginning and reserved for complex cases with multisystem anomalies.
3 - According to the Human Protein Atlas database, there is a relatively high expression of KIT gene in skin. Why the authors did not confirm the effect of non-canonical splice site mutation of c.2484+5G>T in patient’s skin biopsies by RT-PCR or RNA-Seq (whole transcriptome study)?
Response 3: We proposed a skin biopsy to the patients and their legal tutors. However, they denied the authorization. Since this is an invasive procedure, it was not ethical to insist on this request.
4- In silico prediction tools are helpful to classify the variant of unknown significance (VUS) and it is better to have the results of different prediction programs. Therefore, it would be better to add the Combined Annotation-Dependent Depletion (CADD) score of the c.2484+5G>T mutation as well, that is 22.4 for this mutation. CADD is a novel and comprehensive functional annotation tool that the score of 15 can be a cutoff on deleteriousness of a variant. In addition, they can report in silico prediction metrics of the pathogenicity of c.2484+5G>T splice site mutation in Human Splicing Finder (HFS) database.
Response 4: We thank the reviewer for this suggestion. We added the CADD score and HSF3.1 scores. Indeed HSF3.1 also shows the loss of a splicing enhancer, which further corroborates the hypothesis c.2484+5G>T is altering the donor splice site.

Reviewer 2 Report
In the present case report, the authors describe monozygotic twins zygotes with multilocus syndrome in piebaldism: these are my comments:
The phrase “…Skin and hair 45 pigmentation defects represent pathognomonic findings of auditory-pigmentary disorders (APDs),….” is incorrect.
The phrase “..The sequence of the entire coding region of the KIT gene allowed identifying a heterozygous 111 c.2484+5G>T change in intron 17…” is inadequate, coding region or intronic?
It is necessary the analysis, at least, of 100 normal controls to support the “…likely pathogenic.”
In addition of the CGH analysis, exome is necessary due to the variant was classified “,..likely pathogenic.”“…
The pedigree has to be corrected, there are symbologies not described in the family
If only III6 is heterozygous for KIT, what happens with III8-9 and 10
Why think of genetic alterations by paternal line, when the genealogy is by maternal line?
The authors do not present information on PBT gene
No decipher or similar data was used to discard the pathogenic aspect of 15q13.3 microdeletion
No electropherogram of the molecular alteration is shown
The present study requires more laboratory work before being considered for publication
Author Response
We would be honored to resubmit to Medicina the revised manuscript entitled “Congenital sensorineural hearing loss and inborn pigmentary disorders: First report of multilocus syndrome in piebaldism”
Hereinafter, we provide a point-by-point response letter.
English language and style are fine/minor spell check required.
Response: We changed the text as you suggested. English language and style of the manuscript has been completely revised. We have corrected all grammar and spelling errors.
In the present case report, the authors describe monozygotic twins zygotes with multilocus syndrome in piebaldism: these are my comments:
The phrase “…Skin and hair 45 pigmentation defects represent pathognomonic findings of auditory-pigmentary disorders (APDs),….” is incorrect.
Response 1: We have reformulated the sentence.
The phrase “..The sequence of the entire coding region of the KIT gene allowed identifying a heterozygous 111 c.2484+5G>T change in intron 17…” is inadequate, coding region or intronic?
Response 2: We thank the reviewer for identifying this mistake. We Sanger sequenced coding exons and flanking introns. We changed the sentence accordingly.
It is necessary the analysis, at least, of 100 normal controls to support the “…likely pathogenic.” In addition of the CGH analysis, exome is necessary due to the variant was classified “,..likely pathogenic.”“…
Response 3: The analysis of 100 normal controls is no more accepted to exclude that a variant is a polymorphism. Indeed, the recent availability of large databases containing thousands of normal subjects allows a rapid and more careful evaluation of the frequency of the variants found. We did not identify the c.2484+5G>T variant in GnomAD (over 120,000 chromosomes) and added this information in the text as support of a possible pathogenicity. Indeed, the criteria of the American College of Medical Genetics used to classify variants are very restrictive, if compared to the past. Most often a causative variant in a gene only reaches the likely pathogenic threshold, because it lacks functional studies in support for its pathogenicity. Indeed, performing a whole exome sequencing is unlikely to add further data.
The pedigree has to be corrected, there are symbologies not described in the family.
Response 4: We reported in the legend the four possible clinical features present in the different members of the family. Male and female symbols, divided in quadrants, display one or more clinical features up to all four present in the probands.
If only III6 is heterozygous for KIT, what happens with III8-9 and 10
Response 5: Unfortunately, DNA was not available, but we agree with the reviewer it would have been interesting to test them.
Why think of genetic alterations by paternal line, when the genealogy is by maternal line?
Response 6: Our probands have two diseases: one of maternal origin (piebaldism) and one likely paternal (brain function impairment). Indeed, we can only speculate this latter is paternal, because DNA from the father was not available. The father has however a clinics compatible with the 15q13.3 deletion which is a copy number variant associated with a wide expressivity.
The authors do not present information on PBT gene
Response 7: PBT is the acronym of piebaldism.
No decipher or similar data was used to discard the pathogenic aspect of 15q13.3 microdeletion
Response 8: Indeed, the 15q13.3 deletion is a very well-studied pathogenic copy number variant; the available data are so abundant that we have only cited major papers and the OMIM database as comprehensive reference.
No electropherogram of the molecular alteration is shown
Response 9: We have added the electropherograms of the patient and a control. Below we schematized the exon-intron structure and the position of the variant.

Round 2
Reviewer 2 Report
The manuscript is better, however, there are aspects not addressed in the manuscript or at least not indicated, that is, the genealogy continues to show symbology not included (figure 1),
In addition, were normal controls molecularly studied? ,? if not, why..
Author Response
We would be honored to resubmit to Medicina the revised manuscript entitled “Congenital sensorineural hearing loss and inborn pigmentary disorders: First report of multilocus syndrome in piebaldism”
Hereinafter, we provide a point-by-point response letter.
English language and style are fine/minor spell check required.
Answer: We changed the text as you suggested. English language and style of the manuscript has been completely revised. We have corrected all grammar and spelling errors.
The manuscript is better, however, there are aspects not addressed in the manuscript or at least not indicated, that is, the genealogy continues to show symbology not included (figure 1)
Response: we further modified legend to Figure 1, including all symbols used.
In addition, were normal controls molecularly studied? ,? if not, why..
Answer: normal controls for the c.2484+5G>T KIT variant were not tested. This is because the availability of a wide number of controls in the population databases (such as GnomAD), gives a very accurate estimation of the frequency of rare variants. If the c.2484+5G>T is absent from around 120,000 normal chromosomes, adding 200 chromosomes does not increase the information available. The analysis of an adequate number of controls was very popular and mandatory before these databases, but it is now considered useless. The only exception would be the study of a specific population, not represented in normal controls databases, but this is not our case, given the patient is Caucasoid.
